# Antimicrobial and Antibiofilm Activities of *Weissella cibaria* against Pathogens of Upper Respiratory Tract Infections

**DOI:** 10.3390/microorganisms9061181

**Published:** 2021-05-30

**Authors:** Ji-Eun Yeu, Hyeon-Gyu Lee, Geun-Yeong Park, Jisun Lee, Mi-Sun Kang

**Affiliations:** 1R&D Center, OraPharm, Inc., Seoul 04782, Korea; ji-eun85@orapharm.com (J.-E.Y.); gypark@orapharm.com (G.-Y.P.); 2Department of Food and Nutrition, Hanyang University, Seoul 04763, Korea; hyeonlee@hanyang.ac.kr; 3Bio-Healthcare Food Science Interdisciplinary Major, School of Humanities, Art & Technology, Kookmin University, Seoul 02707, Korea; health.livinglab@gmail.com

**Keywords:** probiotic, antimicrobial, antibiofilm, upper respiratory tract, *Weissella cibaria*

## Abstract

Recently discovered preventive effects of probiotics on oral health have attracted interest to their use for the prevention and treatment of various diseases. This study aimed to evaluate the antimicrobial and antibiofilm properties of *Weissella cibaria* against *Streptococcus pyogenes*, *Staphylococcus aureus*, *S*. *pneumoniae*, and *Moraxella catarrhalis*, the major pathogens of upper respiratory tract infections (URTIs). The antimicrobial activities of *W*. *cibaria* were compared with those of other oral probiotics using a competitive inhibition assay and the determination of the minimum inhibitory concentrations (MICs). In addition, a time-kill assay, spectrophotometry, and confocal laser scanning microscopy were used to confirm the antimicrobial and antibiofilm abilities of *W*. *cibaria* CMU (oraCMU) and CMS1 (oraCMS1). Both live cells and cell-free supernatants of all tested probiotics, except *Streptococcus salivarius*, showed excellent antimicrobial activities. All target pathogens were killed within 4 to 24 h at twice the MIC of oraCMU and oraCMS1, which showed the highest antimicrobial activities against *M*. *catarrhalis*. The antimicrobial substances that affected different target pathogens were different. Both oraCMU and oraCMS1 showed excellent abilities to inhibit biofilm formation and remove preformed biofilms. Our results suggest that the *W*. *cibaria* probiotics offer new possibilities for the prevention and treatment of bacterial URTIs.

## 1. Introduction

Upper respiratory tract infections (URTIs) are infections of the nose, sinuses, throat, larynx, and epiglottis [1]. Acute URTIs are primarily caused by viruses [2]. The most common viral URTI is nasopharyngitis (cold), which is experienced by adults 2–5 times a year [3]. Acute URTIs have been reported to cause a high disease burden, accounting for 40% of the reasons for absenteeism among adult workers and for 10% of patients visiting outpatient and emergency rooms [4,5].

Acute URTIs are sometimes caused by bacteria. *Streptococcus pyogenes*, *S*. *pneumoniae*, *Haemophilus influenzae*, *Moraxella catarrhalis*, and *Staph*. *aureus* are common and important pathogens of bacterial infections [6,7,8,9]. In particular, *S*. *pyogenes* is the most clinically important bacterial agent for acute pharyngitis and acute pharyngeal tonsillitis [7].

In children between 6 months and 3 years of age, approximately 90% of acute otitis media are reported to be associated with viral URTIs. In particular, it has been reported that half of the children infected with specific upper respiratory pathogens, such as *S*. *pneumoniae*, *H*. *influenzae*, and *M*. *catarrhalis*, develop acute otitis media after viral URTIs [10].

Although acute URTIs are more frequently associated with viruses than with bacteria, viral infections usually heal spontaneously and rapidly improve. Meanwhile, the rate of improvement of bacterial infections is slow, and the risk of recurrence and chronicity increases as the infection progresses [7,10]. There is no specific treatment other than antibiotics for bacterial URTIs. However, the use of antibiotics has been associated with several problems, including poor patient compliance, allergies, unnecessary side effects, abuse, and tolerance [11,12]. In particular, biofilm-forming bacteria do not respond well to general antibiotic treatments. When biofilms are formed, bacteria have been reported to be 500 times more resistant to antibiotics [13]. Therefore, it is very important to control URTI-related bacteria.

In recent years, there has been increasing interest in the topical application of probiotics to prevent or treat diseases [14]. Several clinical studies have confirmed the effectiveness of probiotics for preventing URTIs. According to Altadill et al. [15], administration of probiotics has beneficial effects on URTIs by significantly reducing the number of days of a URTI and the rate of fever. Another clinical study has reported that the annual episodes of pharyngeal tonsillitis and the incidence of acute otitis media due to streptococcal infection were significantly reduced by the use of probiotics [16]. However, few studies have evaluated the in vitro effectiveness of oral care probiotics in controlling major pathogens of URTIs, which pose a high risk for recurrence and chronicity.

*Weissella cibaria* CMU (oraCMU) and CMS1 (oraCMS1) are oral care live probiotics that are used as commercial strains to aid oral health. These bacteria have been confirmed to be safe in experiments recommended by the Food and Agriculture Organization of the United Nations/World Health Organization and via other animal and human applications [17,18,19]. It has been reported that *W*. *cibaria* has a higher hydrogen peroxide (H_2_O_2_) production potential than that of other commercial oral care probiotics and shows excellent antibacterial and antibiofilm effects against periodontal and dental caries bacteria in the oral cavity [20].

Therefore, in this study, *W*. *cibaria* was evaluated for its antibacterial activity against major pathogens of URTIs, as well as for the ability to inhibit pathogenic biofilm formation and facilitate the removal of biofilms.

## 2. Materials and Methods

### 2.1. Bacterial Strains and Growth Conditions

The four major pathogens of URTIs used in this study were *S*. *pyogenes* KCCM 40411, *Staph*. *aureus* KCTC 1928, *S*. *pneumoniae* ATCC 6303, and *M*. *catarrhalis* KCCM 42707. *S*. *pyogenes* KCCM 40411 and *M*. *catarrhalis* KCCM 42707 were purchased from the Korean Culture Center of Microorganisms (Seoul, Korea). *Staph*. *aureus* KCTC 1928 was purchased from the Korean Collection for Type Cultures (Daejeon, Korea). *S*. *pneumoniae* ATCC 6303 was provided by Chonnam National University (Gwangju, Korea). Five commercial oral care probiotics, including *W*. *cibaria* CMU (oraCMU), *W*. *cibaria* CMS1 (oraCMS1), *Streptococcus salivarius*, *Ligilactobacillus salivarius*, and *Limosilactobacillus reuteri*, were used in this study. The preparations of oraCMU and oraCMS1 were provided by OraPharm, Inc. (Seoul, Korea). *S*. *salivarius* was isolated from a commercial probiotic product using tryptic soy agar (Difco, Detroit, MI, USA). *L*. *salivarius* and *L*. *reuteri* were also isolated from commercial probiotic products using de Man, Rogosa, and Sharpe (MRS) agar (Difco). All bacterial strains were identified using 16S rRNA sequence analysis. *W*. *cibaria*, *L*. *salivarius* and *L*. *reuteri* cultures were grown aerobically in MRS broth at 37 °C for 16 h. *Streptococcus* spp. and *Staph*. *aureus* cultures were grown aerobically in brain heart infusion (BHI) broth (Difco) at 37 °C for 16 h. *Staph*. *aureus* was incubated with shaking, and *S*. *pyogenes* and *S*. *pneumoniae* were incubated under 5% CO_2_ conditions. *M*. *catarrhalis* was cultured on BHI agar plates for 2 days at 37 °C and 5% CO_2_.

### 2.2. Antimicrobial Activity

#### 2.2.1. Competitive Inhibition Assay

A competitive inhibition assay was used to compare the antimicrobial effects of the oral care probiotics on the growth of the target pathogens in coculture. The antimicrobial effects were determined based on the competitive index (CI), which was calculated as follows [21]: CI of the test pathogen = [pathogen colony-forming units (CFU)/probiotic CFU at 16 h]/(pathogen CFU/probiotic CFU at 0 h).

#### 2.2.2. Determination of Minimum Inhibitory Concentrations (MICs)

MICs were determined using a microtiter plate assay [22]. Oral care probiotics were centrifuged at 5000× *g* for 10 min at 4 °C and filtered through a 0.45 μm syringe filter to prepare cell-free supernatants (CFSs). The CFS of each oral care probiotic was serially diluted 2-fold to final concentrations of 31.25 to 500 mg/mL (*v*/*v*). In a 96-well plate, 100 μL of the target pathogen (final concentration: ~5 × 10^6^ CFU/mL) were added to wells containing 100 μL of an oral care probiotic. Each plate included a positive control (target pathogen alone) and negative control (medium). After incubation at each growth condition, the growth of bacteria was determined by measuring the absorbance at 600 nm (OD_600_) using a microplate reader (VersaMax, Molecular Devices, San Jose, CA, USA).

#### 2.2.3. Time-Kill Assay

Time-kill curves [23] were used to evaluate the bactericidal activities of the *W*. *cibaria* strains. CFSs of the *W*. *cibaria* strains were used at MIC and twice the MIC. The target pathogen (final concentration: ~5 × 10^6^ CFU/mL) and each CFS were mixed equally in a 96-well plate. The plates were monitored for bacterial growth over different culture periods at different intervals based on preliminary assay results for each culture condition. Bacterial growth was determined by measuring the number of viable cells and OD_600_.

#### 2.2.4. Characterization of Antimicrobial Substances

The effects of antimicrobial substances, such as organic acids, H_2_O_2_, and bacteriocin-like compounds (BLCs), were evaluated according to our previous study [24]. Briefly, to evaluate the effects of organic acids, CFSs of the *W*. *cibaria* strains were treated with proteinase K (0.1 mg/mL; Sigma-Aldrich, St. Louis, MO, USA) and catalase (0.05 mg/mL; Sigma-Aldrich). After neutralizing the CFS, proteinase K treatment was used to evaluate the effect of H_2_O_2_, and the neutralized CFS was treated with catalase to evaluate the effect of BLCs. The treated samples were serially diluted 2-fold, and 100 μL were added to a 96-well plate. The target pathogen was adjusted to OD_600_ of 0.05 (final concentration: ~5 × 10^6^ CFU/mL) with the growth medium, and 100 μL were inoculated into each well. After incubation of the target pathogen under each condition, OD_600_ was measured using a microplate reader.

### 2.3. Antibiofilm Formation Activity

#### 2.3.1. Inhibition of Biofilm Formation

To investigate the inhibitory abilities of live cells of the *W*. *cibaria* strains on pathogenic biofilm formation, coculture was performed using a Transwell insert (Corning, New York, NY, USA). The target pathogen was inoculated into a well of the lower compartment, and *W*. *cibaria* was inoculated into the upper part of the Transwell insert. The inoculum concentrations of both bacteria were the same (final concentration: ~2.5 × 10^7^ CFU/mL). To evaluate the inhibitory abilities of CFSs of the *W*. *cibaria* strains on pathogenic biofilm formation, the pathogen and CFS were mixed equally in the wells. The target pathogen alone was used as a positive control. The amount of the biofilm formed in each well under each growth condition was measured by spectrophotometry after 24 and 48 h. Briefly, at the end of each incubation period, the culture medium was removed, and after air-drying, each well was stained with 500 µL of 1% crystal violet for 10 min. The stain was removed, and the wells were washed three times with phosphate-buffered saline. One milliliter of absolute ethanol was added to each well to dissolve the stain absorbed by the biofilm. The dissolved stain was dispensed into 96-well plates, and the absorbance was measured at 595 nm using a microplate reader.

#### 2.3.2. Removal of Preformed Biofilms

To investigate whether live cells of the *W*. *cibaria* strains were able to remove preformed pathogenic biofilms, each target pathogen was incubated in the lower chamber of a Transwell insert for 24 h under growth conditions, followed by inoculation of the *W*. *cibaria* strains into the upper chamber of the Transwell insert and incubation for an additional 24 and 48 h. The effects of CFSs of the *W*. *cibaria* strains were evaluated as described above. The target pathogen alone was used as a positive control. The residual amount of biofilm in the well was measured spectrophotometrically, as described above.

#### 2.3.3. Confocal Laser Scanning Microscopy (CLSM) Analysis

To perform CLSM analysis of biofilm formation, a 12 mm diameter coverslip (SPL Life Sciences, Gyeonggi-do, Korea) was placed in a 24-well plate, and culture was performed under the same experimental conditions as described above. The biofilm was gently washed with sterile saline and stained with the Filmtracer LIVE/DEAD biofilm viability kit (Thermo Fisher Diagnostics SpA, Rodano, Italy) according to the manufacturer’s instructions. The stained biofilm was imaged at a 40× magnification using a confocal laser scanning microscope (Leica Microsystems, Wetzlar, Germany). Images were processed and analyzed using Las X (Leica Microsystems CMS GmbH, Mannheim, Germany) to evaluate the overall biofilm volume as an estimate of the total biomass and to calculate the live/dead cell ratio.

### 2.4. Statistical Analysis

The results are presented as the mean ± standard deviation for triplicate measurements. Differences between the means were evaluated using one-way analysis of variance with Duncan’s multiple range test. Differences were considered significant at *p* < 0.05. All statistical tests were performed using SPSS Statistics version 21.0 for Windows (IBM Corp., Armonk, NY, USA).

## 3. Results

### 3.1. Antimicrobial Activity

#### 3.1.1. Competitive Inhibition

CI values were calculated to evaluate competitive inhibition of the major pathogens of URTIs by the oral care probiotics (Figure 1). Except those obtained with *S*. *salivarius*, the CI values of the test pathogens were less than 1. Most of the live probiotics, including the *W*. *cibaria* strains, showed excellent competitiveness against the pathogens, while *S*. *salivarius* had the lowest antimicrobial activity.

#### 3.1.2. MIC Results

The antimicrobial activities of CFSs of the five oral care probiotics against the four major pathogens of URTIs were evaluated using MIC values (Table 1). The best antimicrobial activity against *S*. *pyogenes* was shown by the CFS of *L*. *salivarius* with a MIC of 125 mg/mL. All strains, except *S*. *salivarius*, showed the same MICs of 125 mg/mL for *Staph*. *aureus*, *S*. *pneumoniae*, and *M*. *catarrhalis*. The CFS of *S*. *salivarius* did not show antimicrobial activity against any of the target pathogens.

#### 3.1.3. Time to Kill Major Pathogens of URTIs

Figure 2 shows the killing time for CFSs of the *W*. *cibaria* strains against the four major pathogens. At 2 × MIC, the CFSs of both *W*. *cibaria* strains showed complete bactericidal effects for *S*. *pyogenes* and *M*. *catarrhalis* within 4 and 6 h, respectively, and for *Staph*. *aureus* and *S*. *pneumoniae* within 24 h of exposure.

#### 3.1.4. Characterization of Antimicrobial Substances

The identification of antimicrobial substances of *W*. *cibaria* showed that only organic acids acted on *Staph*. *aureus* and *S*. *pneumoniae*. H_2_O_2_ showed a dose-dependent antimicrobial activity against *S*. *pyogenes* and *M*. *catarrhalis*, and BLCs only acted on *M*. *catarrhalis* (Figure 3).

### 3.2. Antibiofilm Activity

#### 3.2.1. Inhibition of Biofilm Formation

Both live cells and CFSs of the *W*. *cibaria* strains showed similar inhibitory effects on biofilm formation by the four major pathogens of URTIs (Figure 4). After 48 h of incubation, live cells of the *W*. *cibaria* strains significantly reduced biofilm formation by the pathogens (*S*. *pyogenes*, 60–62%; *Staph*. *aureus*, 68–76%; *S*. *pneumonia*, 56–62%; and *M*. *catarrhalis*, 54–59%) (*p* < 0.05). The CFSs of the *W*. *cibaria* strains also significantly reduced biofilm formation by *S*. *pyoge**nes* (80–86%), *Staph*. *aureus* (92–93%), *S*. *pneumoniae* (73–74%), and *M*. *catarrhalis* (56–61%).

#### 3.2.2. Removal of Preformed Biofilms

Both live bacteria and CFSs of the *W*. *cibaria* strains showed similarly excellent biofilm removal abilities for all major pathogens of URTIs, except *S*. *pneumonia* (Figure 5), with the best removal effect observed against the *Staph*. *aureus* biofilm.

#### 3.2.3. CLSM Results

CLSM analysis was used to observe the effects of live cells and CFSs of the *W*. *cibaria* strains on *S*. *pyogenes* biofilm formation. Very small amounts of biofilm were observed in all areas treated with live cells or CFSs of the *W*. *cibaria* strains compared to the positive control group (Figure 6a). In addition, the numbers of dead cells increased in the *W*. *cibaria*-treated groups (Figure 6b and Figure 7).

## 4. Discussion

*W*. *cibaria* is a probiotic found in various fermented foods [25,26,27,28]. In particular, *W*. *cibaria* plays an important role in the maturation of kimchi, a traditional fermented food in Korea, and is a dominant species during the aging process of kimchi. oraCMU and oraCMS1 are *W*. *cibaria* strains isolated and identified from the saliva of 460 children aged 4–7 years with good oral health [29]. These bacteria have been reported to be effective in preventing oral diseases (bad breath, periodontal disease, and tooth decay) by reducing oral pathogens via antibacterial, antibiofilm, and co-aggregation effects [20].

Many previous studies of *W*. *cibaria* have focused on oral health [20,30,31]. We hypothesized that because the upper respiratory tract includes organs that are closest to the oral cavity, *W*. *cibaria* could affect URTIs. Therefore, in this study, *W*. *cibaria* was investigated in vitro to determine its potential for the prevention and treatment of URTIs. *S*. *pneumoniae*, *S*. *pyogenes*, and *M*. *catarrhalis*, the most common bacterial pathogens that cause acute otitis media and pharyngeal tonsillitis, and *Staph*. *aureus*, which is involved in refractory chronic rhinosinusitis [32,33,34], were selected as major pathogens.

In our study, the antimicrobial activities of *W*. *cibaria* strains against the major pathogens of URTIs were compared with those of other commercial oral care probiotics. CIs were calculated to determine the exact nature of the competitive ability of the species. The target pathogens exhibited CIs below 1, which indicated their poor competition in coculture with the oral care probiotics. Except for *S*. *salivarius*, which showed poor competition with *S*. *pyogenes* and *Staph*. *aureus*, all other tested oral care probiotics competed well. The antimicrobial activity of *S*. *salivarius* was the lowest against the four pathogens tested. It was also confirmed that *S*. *pneumoniae* and *M*. *catarrhalis* did not compete well with any oral care probiotic. As a result, except for *S*. *salivarius*, both live cells and CFSs of the other oral care probiotics showed similar and remarkable antimicrobial activities against all pathogens.

*S*. *salivarius* is known to show antimicrobial activity against *S*. *pyogenes*, a sore throat-causing bacterium. However, according to Fiedler et al. [35], *S*. *salivarius* did not kill *S*. *pyogenes* at a concentration similar to that of *S*. *pyogenes*, which was consistent with the failure of *S*. *salivarius* to kill *S*. *pyogenes* in this study. In addition, *S*. *salivarius* did not show any antimicrobial activity against the other pathogens. Meanwhile, the CFS of *L*. *salivarius* showed the best antimicrobial activity, with a MIC of 125 mg/mL for *S*. *pyogenes*, and live bacteria demonstrated excellent competitive inhibition of the pathogens. These findings were consistent with the data of a previous study in which *L*. *salivarius* completely killed *Staph*. *aureus* in coculture [36].

The results of the time-kill assay were useful to determine against which of the four major pathogens of URTIs the bactericidal properties of the *W*. *cibaria* strains were most effective. The results showed that both oraCMU and oraCMS1 were most effective in killing *M*. *catarrhalis* within 4 h at a CFS concentration of 2 × MIC, followed by killing *S*. *pyogenes* within 6 h, while *Staph*. *aureus* and *S*. *pneumoniae* were completely killed within 24 h.

A previous study showed that the CFS of *S*. *pneumoniae* inhibited the growth of *H*. *influenzae*, whereas the CFS of *H*. *influenzae* did not affect the growth of *S*. *pneumoniae* [37]. Furthermore, *H*. *influenzae* was reported to be killed by H_2_O_2_ produced by *S*. *pneumoniae*, and a reversible inhibitory effect of *S*. *pneumoniae* was also observed on the growth of *M*. *catarrhalis*. These data support our findings that *S*. *pneumoniae* was not affected by H_2_O_2_ produced by *W*. *cibaria*, while *M*. *catarrhalis* was affected. In addition, *Staph*. *aureus* was not affected by H_2_O_2_ produced by the *W*. *cibaria* strains, likely because the catalase produced by *Staph*. *aureus* [38] eliminated the effects of H_2_O_2_. Therefore, it was confirmed that the *W*. *cibaria* strains had stronger antimicrobial activities when antimicrobial substances such as organic acids, H_2_O_2_, and BLCs acted together, as was observed in the case of *M*. *catarrhalis*.

Biofilm formation, which is due to the secretion of extracellular polymeric materials, provides stability to microbial populations by allowing bacteria to adhere to surfaces. In this state, an initial biofilm is formed when bacterial cells form a colony. In the process of biofilm growth and maturation, bacterial cells are released, and free-floating bacteria are disseminated to cause infection in surrounding tissues or other organs [39].

Biofilms ensure good survival and protection of pathogens from host defense mechanisms, antibiotics, and other environmental factors [40]. Therefore, once a biofilm is formed, it is difficult to eradicate, even with conventional antibiotic treatment [41]. Moreover, it is well known that the use of antibiotics not only does not reduce complications of bacterial infections, but increases medical costs by causing side effects and antibiotic resistance [3,4,5]. Biofilm formation by pathogens of URTIs can be an important cause of chronic infectious diseases of the upper respiratory tract, including recurrent middle ear disease, chronic rhinosinusitis, and recurrent pharyngeal tonsillitis [41]. Therefore, it is most effective to prevent pathogens from forming a biofilm, for which new treatments are needed.

In the present study, the *W*. *cibaria* strains were tested for their ability to affect biofilm formation by the four major pathogens of URTIs. In a Transwell assay, in which live cells of the *W*. *cibaria* strains were not directly cultured with the target pathogen, inhibition of pathogenic biofilm formation was observed. In addition, both live cells and CFSs of the *W*. *cibaria* strains showed the ability to facilitate biofilm removal, although they were more effective in inhibiting biofilm formation than in removing preformed biofilms. These results suggest that the antibiofilm activities of the *W*. *cibaria* strains are due to secreted substances and do not require direct contact with pathogens.

Similar to the results of Bidossi et al. [42], *W*. *cibaria* strains inhibited biofilm formation by *S*. *pneumoniae*, *M*. *catarrhalis*, and *Staph*. *aureus*. In addition, biofilm formation by *S*. *pyogenes*, which is not significantly affected by the presence of probiotic strains, was also inhibited, and the biofilm was removed. Wang et al. [43] reported that *W*. *cibaria* FbpA prevented *Staph*. *aureus* colonization by interfering with the invasion pathway and inhibiting biofilm formation. Similarly, in our study, the *W*. *cibaria* strains were most effective in inhibiting biofilm formation and removing biofilm of *Staph*. *aureus*, among the four target pathogens.

URTIs are mainly caused by viruses [2], but the prevention and treatment of infections caused by certain bacterial pathogens are also important. To the best of our knowledge, this study is the first to confirm the preventive effects of *W*. *cibaria* on URTIs in vitro. The results indicated that both *W*. *cibaria* probiotic strains had desirable functions to be used against pathogens of URTIs.

Substances that can regulate the expression of protein mediators of inflammatory responses and signaling pathways are known to play an important role in the prevention and treatment of URTIs [14,44]; therefore, further studies are needed.

## 5. Conclusions

This study reports the antimicrobial activities of *W*. *cibaria* oral care probiotics against major pathogens of URTIs, including the inhibition of biofilm formation. These findings suggest the potential of *W*. *cibaria* to be used as an alternative prevention and treatment agent in the management of URTIs.

## Figures and Tables

**Figure 1 microorganisms-09-01181-f001:**
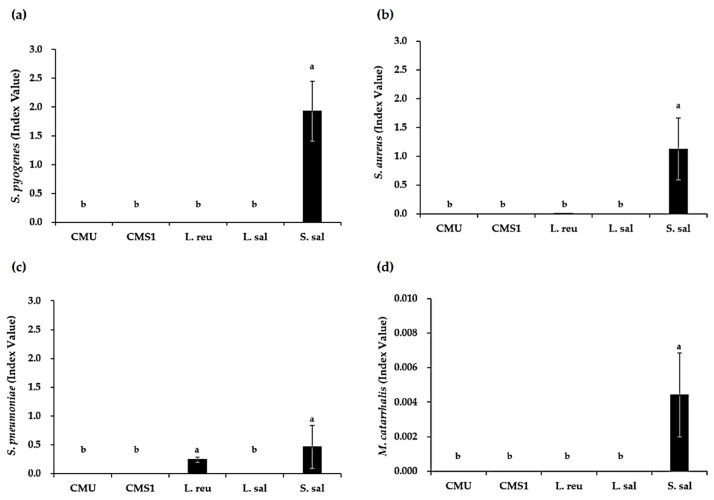
Competitive indexes (CIs) in coculture of oral care probiotics and major pathogens of URTIs. (**a**) *Streptococcus pyogenes*, (**b**) *Staphylococcus aureus*, (**c**) *Streptococcus pneumoniae*, and (**d**) *Moraxella catarrhalis*. Different letters (a and b) indicate significant differences at *p* < 0.05. CMU, *Weissella cibaria* CMU; CMS1, *W*. *cibaria* CMS1; L. reu, *Limosilactobacillus reuteri*; L. sal, *Ligilactobacillus salivarius*; S. sal, *Streptococcus salivarius*.

**Figure 2 microorganisms-09-01181-f002:**
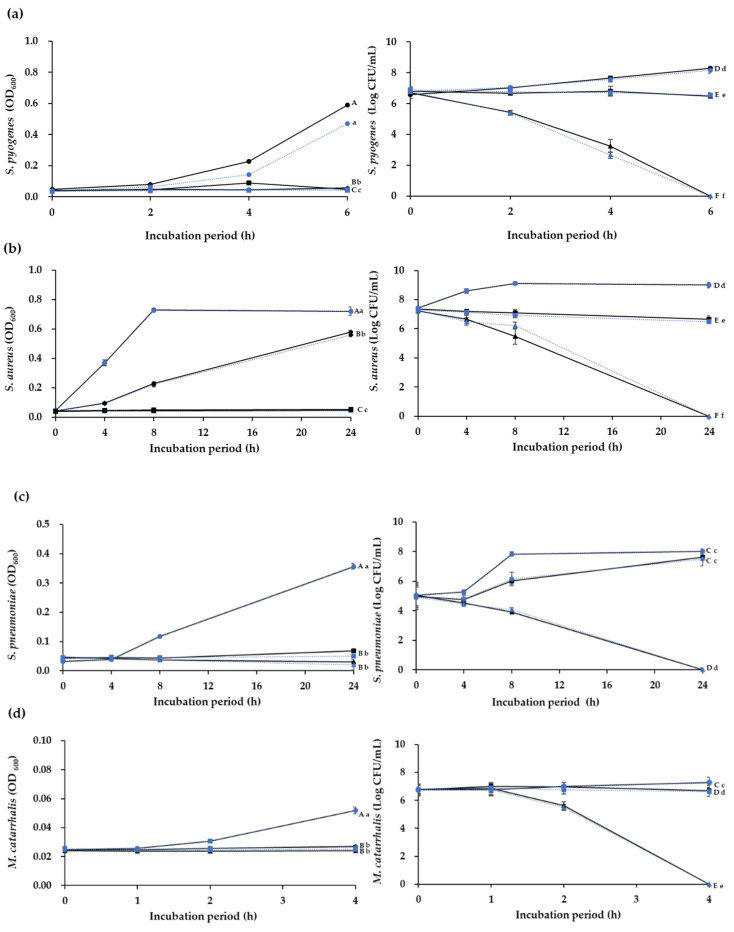
Time-kill curves for cell-free supernatants of *Weissella cibaria* strains against (**a**) *Streptococcus pyogenes*, (**b**) *Staphylococcus aureus*, (**c**) *S*. *pneumoniae*, and (**d**) *Moraxella catarrhalis*, at different minimum inhibitory concentration (MIC) increments: ▲, 2 × MIC; ■, 1 × MIC; ●, untreated control. Solid line, *W*. *cibaria* CMU (oraCMU); broken line, *W*. *cibaria* CMS1 (oraCMS1). Different letters (A–F) indicate significant differences among oraCMU treatment groups (*p* < 0.05). Different letters (a–f) indicate significant differences among oraCMS1 treatment groups (*p* < 0.05). OD_600_, absorbance at 600 nm; CFU, colony-forming units.

**Figure 3 microorganisms-09-01181-f003:**
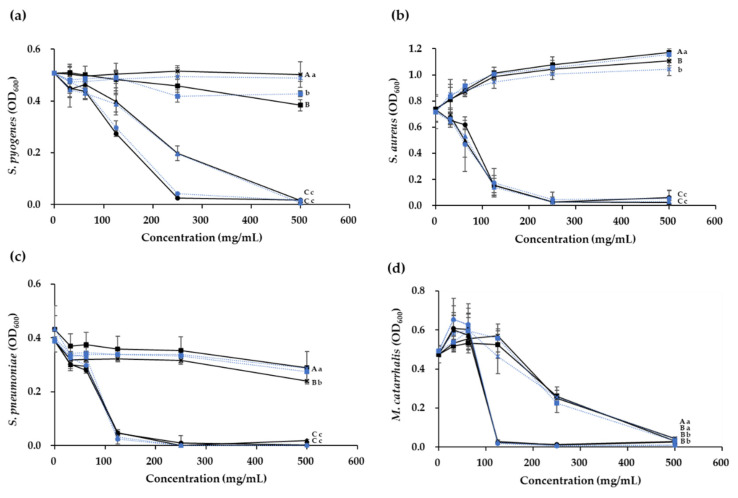
Dose-dependent effects of organic acids, H_2_O_2_, and bacteriocin-like compounds (BLCs) in cell-free supernatants (CFSs) of *W**eissella*
*cibaria* strains against (**a**) *Streptococcus pyogenes*, (**b**) *Staphylococcus aureus*, (**c**) *S*. *pneumoniae*, and (**d**) *Moraxella catarrhalis*. ●, CFS; ▲, organic acids; ■, H_2_O_2_; X, BLCs. Solid line, *W*. *cibaria* CMU (oraCMU); broken line, *W*. *cibaria* CMS1 (oraCMS1). Different letters (A–C) indicate significant differences among oraCMU treatment groups (*p* < 0.05). Different letters (a–c) indicate significant differences among oraCMS1 treatment groups (*p* < 0.05). OD_600_, absorbance at 600 nm.

**Figure 4 microorganisms-09-01181-f004:**
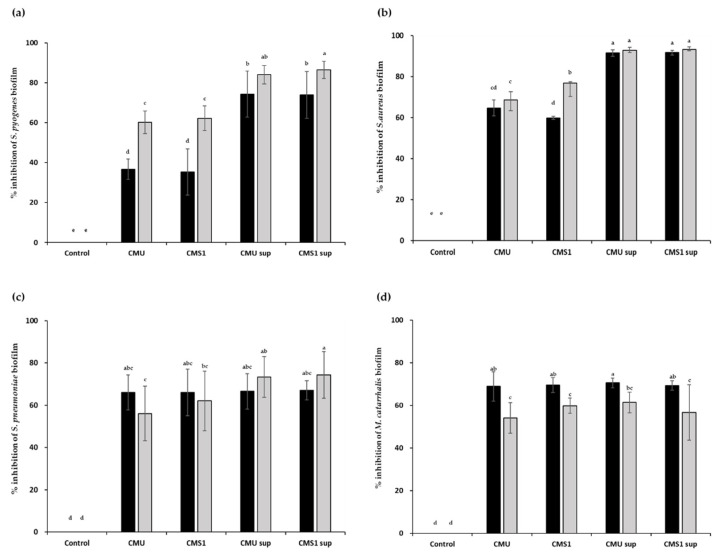
Inhibitory effects of *Weissella cibaria* strains on biofilm formation by (**a**) *Streptococcus pyogenes*, (**b**) *Staphylococcus aureus*, (**c**) *S*. *pneumoniae*, and (**d**) *Moraxella catarrhalis*. Black bar, 24 h; gray bar, 48 h. CMU, *W*. *cibaria* CMU; CMS1, *W*. *cibaria* CMS1; CMU sup, cell-free supernatant (CFS) of *W*. *cibaria* CMU; CMS1 sup, CFS of *W*. *cibaria* CMS1. Different letters (a–e) indicate significant differences at *p* < 0.05.

**Figure 5 microorganisms-09-01181-f005:**
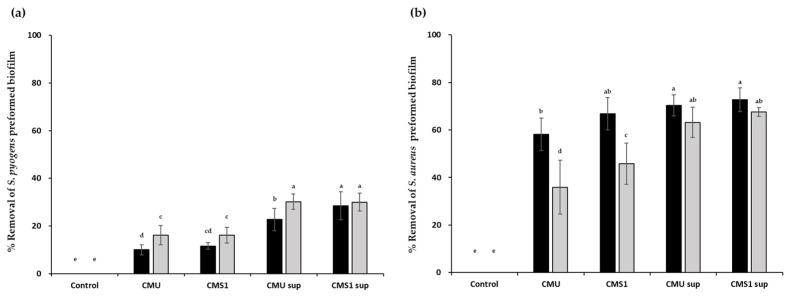
Removal effects of *Weissella cibaria* strains on preformed biofilms of (**a**) *Streptococcus pyogenes*, (**b**) *Staphylococcus aureus*, (**c**) *S*. *pneumoniae*, and (**d**) *Moraxella catarrhalis*. Black bar, 24 h; gray bar, 48 h. CMU, *W*. *cibaria* CMU; CMS1, *W*. *cibaria* CMS1; CMU sup, cell-free supernatant (CFS) of *W*. *cibaria* CMU; CMS1 sup, CFS of *W*. *cibaria* CMS1. Different letters (a–e) indicate significant differences at *p* < 0.05.

**Figure 6 microorganisms-09-01181-f006:**
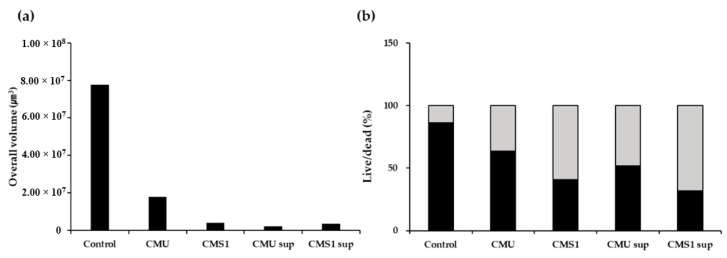
Results of confocal scanning microscopy analysis. (**a**) Overall biofilm volumes; (**b**) live/dead cell ratios. Black bar, live cells; gray bar, dead cells. CMU, *W**eissella cibaria* CMU; CMS1, *W*. *cibaria* CMS1; CMU sup, cell-free supernatant (CFS) of *W*. *cibaria* CMU; CMS1 sup, CFS of *W*. *cibaria* CMS1.

**Figure 7 microorganisms-09-01181-f007:**
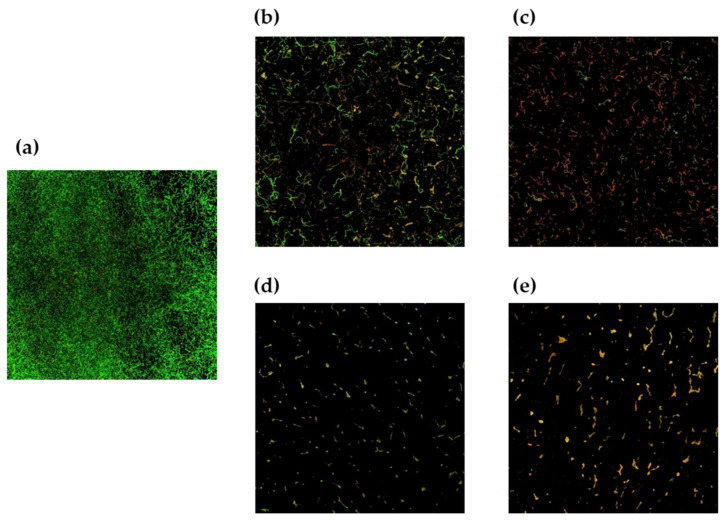
Confocal laser scanning microscopy images of *S**treptococcus pyogenes* biofilms. (**a**) Untreated control; (**b**) treatment with live cells of *W**eissella*
*cibaria* CMU (oraCMU); (**c**) treatment with live cells of *W*. *cibaria* CMS1 (oraCMS1); (**d**) treatment with a cell-free supernatant (CFS) of oraCMU; (**e**) treatment with a CFS of oraCMS1. Green, live cells; red, dead cells (magnification, 40×).

**Table 1 microorganisms-09-01181-t001:** Minimum inhibitory concentrations (MICs) of cell-free supernatants of oral care probiotics against pathogens.

Probiotic	MIC (mg/mL)			
	*S*. *pyogenes*	*Staph*. *aureus*	*S*. *pneumoniae*	*M*. *catarrhalis*
*W. cibaria* CMU	250	125	125	125
*W. cibaria* CMS1	250	125	125	125
*L. reuteri*	250	125	125	125
*L. salivarius*	125	125	125	125
*S. salivarius*	>500	>500	>500	>500

*S. pyogenes*, Streptococcus pyogenes; *Staph. aureus*, Staphylococcus aureus; *S. pneumoniae*, Streptococcus pneumoniae; *M. catarrhalis*, Moraxella catarrhalis; *W. cibaria* CMU, Weissella cibaria CMU; *W. cibaria* CMS1, Weissella cibaria CMS1; *L. reuteri*, Limosilactobacillus reuteri; *L. salivarius*, Ligilactobacillus salivarius; *S. salivarius*, Streptococcus salivarius.

## Data Availability

The data presented in this study are available on request from the corresponding author.

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
