# Peer review of "Antimicrobial and Antibiofilm Activities of Weissella cibaria against Pathogens of Upper Respiratory Tract Infections"

_microorganisms, 2021, doi:10.3390/microorganisms9061181_

Round 1

Reviewer 1 Report

A very interesting manuscript on the effect of Weissella cibaria strains against pathogens of URTI. The study is well-designed, the materials and methods are appropriate for the study, the results adequately described and discussed. Only a few minor suggestions can be offered:

l.14, 35, 204, 245. abbreviation can be used for the second ‘Streptococcus’. Throughout the text, the abbreviation of Staphylococcus cannot be the same with the respective of Streptococcus.

l. 84, 185. please use the new lactobacilli nomenclature; i.e. Limosilactobacillus reuteri instead of Lactobacillus reuteri and Ligilactobacillus salivarius instead of Lactobacillus salivarius. Abbreviations can be arranged accordingly. word ‘Lactobacillus spp.’ (l. 89) has to be accordingly modified.

Table 1. abbreviations of the genera can be used and explained in a footnote

Figure 4 (d) last column on the right (light grey CMS1 sup). please verify ‘c’ because it looks weird next to the ‘ab’ of the previous column

Author Response

Response to Reviewer 1 Comments

A very interesting manuscript on the effect of Weissella cibaria strains against pathogens of URTI. The study is well-designed, the materials and methods are appropriate for the study, the results adequately described and discussed. Only a few minor suggestions can be offered

  1. 14, 35, 204, 205. Abbreviation can be used for the second ‘Streptococcus’.

Throughout the text, the abbreviation of Staphylococcus cannot be the same with the respective of Streptococcus.

Response: We appreciate your valuable suggestion, and have revised the text as suggested.

  1. 84, 185. Please use the new lactobacilli nomenclature; i.e. Limosilactobacillus reuteri instead of Lactobacillus reuteri and Ligilactobacillus salivarius instead of Lactobacillus salivarius. Abbreviations can be arranged accordingly. word ‘Lactobacillus spp.’ (l. 89) has to be accordingly modified.

Response: We appreciate your valuable suggestion, and have revised the sentence as suggested.

Table 1. abbreviations of the genera can be used and explained in a footnote.

Response: Thank you for your comments. We have revised the text as suggested.

Figure 4 (d) last column on the right (light grey CMS1 sup). Please verify ‘c’ because it looks weird next to the ‘ab’ of the previous column.

Response: As a result of statistical analysis, it was confirmed that there was a statistically significant difference, and there were no abnormalities.

Reviewer 2 Report

The article aimed to evaluate Weissella cibaria antimicrobial and antibiofilm properties against pathogenic strains responsible of upper-respiratory tract infections (URTIs). The authors compared weissela cibaria properties to oral commercial probiotic strains. Weissela cibaria cells as well medium culture supernatant showed antimicrobial and antibiofilm properties against pathogen strains.

  • In the aim to use thus bacteria as probiotic to fight URTIs, how do you consider the probiotic administration, it will be by spray or other mode of administration?
  • Weillsela cibaria is present in fermented food as kimchi, do you know is there is meta-analysis which may correlate kimchi consumption and a low URTIs occurence rate?
  • For figure 2 and figure 3, I suggest to simplify results presentation to make results more readable from graphics, you can use colors for figure 3 or presented only two MIC concentration for figure 2.
